# Influence of n-ZnO Morphology on Sulfur Crosslinking and Properties of Styrene-Butadiene Rubber Vulcanizates

**DOI:** 10.3390/polym13071040

**Published:** 2021-03-26

**Authors:** Dariusz M. Bieliński, Katarzyna Klajn, Tomasz Gozdek, Rafał Kruszyński, Marcin Świątkowski

**Affiliations:** 1Institute of Polymer & Dye Technology, Lodz University of Technology, 90-924 Lodz, Poland; katarzyna.klajn@dokt.p.lodz.pl (K.K.); tomasz.gozdek@p.lodz.pl (T.G.); 2Institute of General and Ecological Chemistry, Lodz University of Technology, 90-924 Lodz, Poland; rafal.kruszynski@p.lodz.pl (R.K.); marcin.swiatkowski@p.lodz.pl (M.Ś.)

**Keywords:** n-ZnO, rubber, vulcanization, activation energy, crosslink structure, mechanical properties

## Abstract

This paper examines the influence of the morphology of zinc oxide nanoparticles (n-ZnO) on the activation energy, vulcanization parameters, crosslink density, crosslink structure, and mechanical properties in the extension of the sulfur vulcanizates of styrene-butadiene rubber (SBR). Scanning electron microscopy was used to determine the particle size distribution and morphology, whereas the specific surface area (SSA) and squalene wettability of the n-ZnO nanoparticles were adequately evaluated using the Brunauer–Emmet–Teller (BET) equation and tensiometry. The n-ZnO were then added to the SBR in conventional (CV) or efficient (EV) vulcanization systems. The vulcametric curves were plotted, from which the cure rate index (CRI) rate of the vulcanization and the activation energy were calculated. The influence on the mechanical properties of the SBR vulcanizates was stronger in the case of the EV curing system than when the CV curing system was used. Of the vulcanizates produced in the EV curing system, the best performance was detected for n-ZnO particles with a hybrid morphology (flat-ended rod-like particles on a “cauliflower” base) and high SSA. Vulcanizates produced using the CV curing system showed slightly better mechanical properties after the addition of nanoparticles with a “cauliflower” morphology than when the rod-like type were used, irrespective of their SSA. In general, nanoparticles with a rod-like structure reduced the activation energy and increased the speed of vulcanization, whereas the cauliflower type slowed the rate of the process and the vulcanizates required a higher activation energy, especially when using the EV system. The crosslink structures were also more clearly modified, as manifested by a reduction in the polysulfidic crosslink content, especially when n-ZnO activators with a rod-like morphology were applied.

## 1. Introduction

Vulcanization is the process of forming a three-dimensional network structure using a linear polymer via chemical reactions involving sulfur [1]. The formation of a network structure is crucial for obtaining rubber vulcanizates with elastomeric properties. The various reagents associated with the sulfur vulcanization of polydienes, such as styrene-butadiene rubber (SBR), include vulcanization agents, accelerators, activators, retarders, and prevulcanization inhibitors. Vulcanization agents include elemental sulfur and/or an organic sulfur donor, such as tetramethylthiuram disulfide (TMTD). The most important classes of accelerators are those based on sulfenamides, benzothiazoles, guanadines, or dithiocarbamic acid (Table 1).

Rubber vulcanization using sulfur without accelerators takes several hours and is not commercially viable. The application of accelerators makes the process significantly faster, enabling curing within a period of several minutes. Sometimes known as secondary accelerators, such activators include metal oxides (usually zinc oxide (ZnO)), fatty acids (usually stearic acid), and nitrogen-containing bases. Retarders and prevulcanization inhibitors may also be added so as to prolong the processing times and avoid premature vulcanization (scorch). A typical basic recipe for accelerated sulfur vulcanization includes sulfur and/or a sulfur donor (0.5–4 phr), an accelerator or a mixture of accelerators, ZnO (2–10 phr), and stearic acid (1–4 phr). The type and the amount of the sulfur donor and the accelerator are the major variables. Vulcanization systems are classified as conventional (CV), semi-efficient (semi-EV), or efficient (EV), depending on the level of sulfur and the ratio of accelerator to sulfur (*A*/*S*), as shown in Table 2.

Rubber vulcanizates show various levels of crosslinking efficiency and develop different crosslink structures, depending on the curing system used (Table 3).

The energy, length, and stiffness of the crosslinks together determine the mechanical and thermal properties of rubber vulcanizates [3]. ZnO acts as an activator of sulfur vulcanization, forming zinc-based complexes during the first steps of the reaction. These zinc-based complexes have a major influence on both the kinetics of the reaction and the nature of the crosslink product [4]. However, its low affinity towards rubber requires the use of large amounts (3–5 phr) of ZnO in order to achieve a good distribution in the matrix. Various possible ways of reducing the amount of ZnO needed in the process of rubber compounding have been proposed [5]. Different zinc complexes have been tested as alternatives to the conventional ZnO and fatty acid activator system [6]. Of particular interest is nano zinc oxide (n-ZnO). Its nanoscale particle distribution could potentially increase the accessibility of zinc ions to curing agents. Cure characteristics suggest that the use of n-ZnO can reduce the consumption of zinc by a factor of 10 [7].

Many studies have focused on the synthesis of n-ZnO. However, there has been little research into how the morphology of n-ZnO particles might influence vulcanization. Przybyszewska and Zaborski [8] studied the effect of the size (in terms of surface area) and shape (spheres, whiskers, or snowflakes) of ZnO particles on the crosslink density and mechanical behavior of carboxylated nitrile rubber. The rubber vulcanizates prepared with n-ZnO showed a higher crosslink density and improved mechanical properties compared with the vulcanizates obtained using conventional ZnO. However, no specific trends were found correlating the particle size and specific surface area of the n-ZnO particles with the crosslink efficiency. Panampilly et al. [9] studied the effect of nanosized ZnO on NR vulcanization. Even with a very small amount of n-ZnO (0.5 phr), they observed a reduction in the optimum cure time (t90) and a higher cure rate index (CRI). These findings were attributed to an increase in rubber–filler interactions, as confirmed by the higher bound rubber (BdR) content [10]. Likewise, Roy et al. [11] found that the addition of 0.5 phr of n-ZnO led to a decrease in t90 and a higher cure rate index (CRI), compared with the addition of 5 phr of conventional ZnO. A thermo gravimetric analysis (TGA) revealed that the rubber vulcanizates containing n-ZnO had a better thermal stability because of a decrease in the thermal motion of the polymer chains within the network structure. Cui et al. [12] analyzed the efficiency of using n-ZnO for the vulcanization of SBR in two different structural forms, but with approximately the same particle size and specific surface area—“table-like” (T-ZnO) and “rod-like” nanosized ZnO (R-ZnO). The curing rate was faster in the vulcanization system with R-ZnO and the vulcanizates also showed both higher crosslink densities and better mechanical properties compared with those prepared with T-ZnO. The differences in the behavior of the n-ZnO structures were attributed to the larger amounts of free Zn^2+^ ions that formed in the system with R-ZnO.

In summary, according to the literature, reducing the size of ZnO particles can be an effective strategy for enhancing the vulcanization efficiency. Nanometric dimensions favor a better dispersion and distribution of the activator in the polymer matrix, because of the higher surface area of n-ZnO. This promotes the interaction of the zinc centers with the curing agents and rubber chains. The aim of the present research was to increase our understanding of the effect of the shape of n-ZnO particles on the curing activity. We also examined the influence of the particle shape on the crosslink structures of SBR vulcanizates cured with sulfur systems of varying efficiencies.

## 2. Materials and Methods

### 2.1. Materials

#### 2.1.1. Synthesis and Characterization of n-ZnO

The n-ZnO nanoparticles used in this study were synthesized according to procedures described in detail in the literature [13,14]. The precursors and key synthesis features are summarized in Table 4.

The synthetized n-ZnO samples were characterized using scanning electron microscopy (SEM) to determine their morphology, size, and shape. The samples were also studied to determine their specific surface area (SSA) and squalene wettability (squalene was used as a model of rubber [15]). For a description of the experimental procedures, see Section 2.2.

#### 2.1.2. Preparation of Rubber Vulcanizates

The compositions of the studied rubber compounds are presented in Table 5.

Designation of samples:CV/EV—rubber compounds containing n-ZnO in a conventional/effective curing system;R—reference rubber compound containing standard ZnO;B, C, D, E and G—rubber compounds containing a specific kind of n-ZnO.

All of the ingredients were mixed in a Brabender-Plasticorder Lab-Station (Brabender GmbH & Co. KG, Duisburg, Germany) laboratory 80 cm^3^ mixer in three steps, according to the PN-ISO 2393:2015-12 standard. They were then sheeted using a David Bridge 2 Roll Mill (David Bridge & Co., Rochdale, U.K.) laboratory two-roll mill. Samples of the rubber vulcanizates were prepared by steel molding using a laboratory hydraulic press operating under a pressure of 200 bar and at a temperature of 160 °C, for an optimal vulcanization time (*t*_90_) determined rheometrically by a MDR 2000 vulcameter (Alpha Technology, Hudson, OH, USA), according to PN-ISO:6502.

### 2.2. Methods

#### 2.2.1. Scanning Electron Microscopy (SEM)

The morphology of the standard (R) and the synthetized n-ZnO particles was analyzed using a Hitachi S-4700 FE-SEM (Hitachi, Krefeld, Germany) operating with a BSE detector under low vacuum conditions. The samples of n-ZnO were placed on the adhesive carbon discs, without any coating. The size of nanoparticles were determined from SEM images (for samples B, C and D, it was their diameter).

#### 2.2.2. Specific Surface Area (BET)

The specific surface areas (SSA) of the standard (R) and the synthetized n-ZnO powders were determined using an ASAP 2010 instrument (Micromeritics, Norcross, GA, USA), applying the Brunauer-Emmet-Teller (BET) equation. The SSA values were determined by applying the five-point BET procedure, according to ASTM D3037 and ASTM D4820 standards.

#### 2.2.3. Squalene Wettability

The wettability of the synthetized n-ZnO powders with squalene was determined using a Krűss K 100 tensiometer (Krűss GmbH, Hamburg, Germany). Glass capillaries containing about 0.5 g of the n-ZnO samples were placed in the apparatus and were tested using the sorption method. The slopes of the squalene sorption lines for the different n-ZnO particles were determined and compared.

#### 2.2.4. Kinetics of Vulcanization

The vulcanization kinetics of the rubber compounds were analyzed using an Alpha Technologies MDR 2000 oscillating disk rheometer (Alpha Technologies, Hudson, OH, USA) at 150 °C, according to PN-ISO: 6502. The following curing parameters were determined based on the experimental data: optimal vulcanization time (*t*_90_), vulcanization scorch time (*t*_s2_), max. torque (*MH*), and min. (*ML*) torque. The increase of torque was calculated as Δ*M* = *MH* − *ML*. The conventional cure rate index (*CRI*) of the rubber compounds was calculated according to Equation (1) [16,17], as follows:(1)$$CRI\;=\;\frac{100}{t_{90}-\;t_{s2}}$$

Based on the vulcametric data, the vulcanization kinetics of the rubber mixes were characterized in terms of the activation energy, calculated according to the Arrhenius Formula (2):ln *k*(T) = ln*A* − *E*_a_/(*RT*)(2)

The rate constant (k) was calculated using non-linear regression according to the Kamal–Sourour [18] model (3):*dα*/*dt* = 1/(*M*_H_ − *M*_L_) × *dM*/*dt*(3)
where *α*(t) is the degree of vulcanization at a given time and *M*(t) is the torque at a given time. This enabled the course of the vulcanization speed (*d*α/*d*t) to be determined as a function of the degree of vulcanization (*α*).

#### 2.2.5. Crosslink Density and Structure

The equilibrium swelling of the rubber vulcanizates in the toluene was determined according to the standard procedure, as described in the literature [19]. The crosslink density of the vulcanizates was calculated from their volumetric equilibrium swelling values, applying the Flory–Rehner equation [20] with an SBR–toluene Flory–Huggins parameter of 0.378 [21]. The structural composition of the sulfidic crosslinks was evaluated according to the procedure described by Saville and Watson [22], based on selective dissolving in thiol–amine solvents (OTAM/OTAT), depending on the crosslink length.

#### 2.2.6. Mechanical Properties of Rubber Vulcanizates

The mechanical properties of the rubber vulcanizates were determined using a Zwick 1435 universal mechanical testing machine (ZwickRoell GmbH & Co. KG, Ulm, Germany), according to ISO 37. Dumbbell-shaped specimens of around 1.5 mm thickness were used. Five samples for each material were analyzed.

## 3. Results & Discussion

### 3.1. Characterization of n-ZnO Particles

Figure 1 shows SEM images of the synthetized n-ZnO particles, together with their specific surface areas (SSA). An image of the standard ZnO (R) is provided for the purposes of comparison.

The standard ZnO sample (R) showed a high degree of agglomeration and a low SSA. The agglomerates were comparable in shape to the clusters of rods or balls stuck together.

The nanometric ZnO samples were classified into three groups, based on their morphologies. Types B and C had fairly high length to width ratios. Type B was composed of sharp-ended rod-like particles (needles), arranged quite close to one another, whereas type C was the bigger, flat-ended, rod-like particles with greater distances between them. Types E and G resembled the reference sample, in that they had a cauliflower-shaped structure. However, type G had a much smaller native particle size and much higher SSA in comparison with type E, and both types had a significantly smaller native particle size than the reference. Type D was a hybrid combination of shapes from types G and C, composed of flat-ended rod-like particles emerging from a “cauliflower” base. Importantly, the SSA depended not only on the size of individual particles, but also on their arrangement among themselves, which may have affected their wettability and dispersion in rubber [23].

Table 6 compares the squalene wettability of the synthetized n-ZnO particles. The results for standard ZnO are presented for the purposes of comparison.

Nanoparticles of the D and B types exhibited a significantly lower wettability by squalene in comparison with the standard ZnO particles, despite their higher SSA. Probably because of their very high SSA, the C-type n-ZnO particles showed similar wettability to the reference. Rather surprisingly, the “cauliflower” morphology of types E and G showed the highest wettability by squalene. Even the type E nanoparticles with a relatively low SSA showed a significantly enhanced wettability in comparison with the reference ZnO particles. It can be concluded that the morphology of n-ZnO particles played a more important role in determining the wettability by squalene than the SSA. The cauliflower morphology was found to be superior to the rod-like morphology in terms of the wetting efficiency. However, the effect of SSA appeared to be more pronounced in the case of rod-like nanoparticles. This shows that the wettability was dependent on SSA, which is in agreement with other studies on filler activity [24]. In terms of its potential interactions with rubber macromolecules, the n-ZnO particles with a rod-like morphology seemed to be less effective than the reference ZnO, irrespective of their SSA, shape, and packing density. It seems likely that rod-like particles emerging from a cauliflower base adversely affected the efficient “cauliflower” morphology, which seemed much more prone to wetting by rubber macromolecules than the other n-ZnO particles studied.

### 3.2. Kinetics of Rubber Vulcanization

#### 3.2.1. Conventional Vulcanization System

As expected, the rubber compounds containing n-ZnO began to vulcanize earlier and finished the process faster, resulting in a greater increase in the vulcametric modulus (Δ*M*) compared with the system containing standard ZnO (CV_R) [25]. These results are presented in Table 7.

The lowest optimal time of vulcanization was observed in the case of rubber mixes containing n-ZnO particles with a cauliflower morphology (E and G), irrespective of the SSA. Their curing rate index was also higher in comparison with the reference compound containing the standard ZnO, even in the case of type E zinc oxide particles with a low SSA. However, as shown in Figure 2, the most rapid rate of vulcanization as a function of the vulcanization degree dα/dt was observed with the use of rod-like n-ZnO particles (types C and B) or with particles of a hybrid morphology (D). The cauliflower-shaped nanoactivators (G and E) behaved similarly to the reference ZnO (R).

As shown in Figure 3, the cauliflower-shaped particles exhibited the highest activation energy of the studied n-ZnO samples during vulcanization, although it was of course lower than that of the micrometric activator (R). The use of rod-shaped n-ZnO particles appears to have increased the speed of sulfur CV vulcanization, although the average values for CRI were lower. Generally, replacing the standard ZnO with nanometric ZnO led to an increase in the vulcanization yield. The lower Δ*M* value in the case of the D-type n-ZnO was either the result of a lower crosslink density or a consequence of changes to the crosslink structure [26].

#### 3.2.2. Effective Vulcanization System

The use of the EV curing system resulted in a smaller increase in the vulcametric modulus of the rubber compounds, compared with the results for the CV system (Table 8). Again, the application of n-ZnO activators caused the process of vulcanization to start earlier. This was especially visible in the case of EV_D. However, it did not finally lead to a higher vulcametric modulus compared with the reference curing system (EV_R).

As had been observed in the CV sulfur curing system, the EV systems also showed faster vulcanization when cauliflower-shaped activators (especially type G with high SSA) or particles with a hybrid morphology were used. However, in the EV systems, the vulcanization reaction rate was related to the SSA of the activator—the speed of the reaction was the highest for types G and D. In the EV systems, contrary to the experimental data for the CV curing systems, dα/dt followed the order of the calculated CRI values (Figure 4). The only exception was the sample with cauliflower-shaped n-ZnO activators (E) with a low SSA, which were associated with a high speed of vulcanization (similar to that for D) but a moderate CRI value.

Importantly, the nanoparticles with a rod-like morphology reduced the activation energy much more than any of the other n-ZnO samples, although this effect was not associated with either lower ts_2_ or with lower CRI values. Rather unexpectedly, the n-ZnO particles with a cauliflower shape (E and G) or hybrid morphology (D) were less effective in the EV curing system, which showed higher activation energy compared to the system with standard ZnO (EV-R). The results are presented in Figure 5.

### 3.3. Crosslink Density and Structure of Rubber Vulcanizates

#### 3.3.1. Conventional Vulcanization System

The crosslink densities of the rubber compounds prepared in the CV system were uncorrelated with the SSA of the n-ZnO activators used (Figure 6A). The morphology and SSA likely affected the dispersion n-ZnO in the rubber. Nanoparticles agglomerated, reduced their reactivity, and resulted in no correlation between the SSA and crosslink density [27].

The vulcanizates containing the type G nanoparticles showed the highest crosslink density. The vulcanizates containing the B-type n-ZnO also had a high crosslink density. Although the Δ*M* data might lead us to expect that the vulcanizates containing type E nanoparticles would exhibit high crosslink densities, in fact they had the lowest crosslink density. Figure 6A shows the differences between the crosslink densities of the studied vulcanizates, and Figure 6B presents the differences between the crosslink structures. In contrast to the CV_R samples, the curing systems containing n-ZnO were found to have similar amounts of C–C and monosulfidic crosslinks, smaller amounts of disulfidic crosslinks, and higher amounts of polysulfidic crosslinks. This suggests that the nanoactivator partially prevented the maturation of the network [28]. There was very little difference in terms of the crosslink structure between the vulcanizates containing n-ZnO. However, the vulcanizates containing the cauliflower-shaped n-ZnO showed a slightly lower content of C–C and monosulfidic crosslinks.

#### 3.3.2. Effective Vulcanization System

As was the case with the rubber compounds produced in the CV system, the crosslink densities of the rubber compounds produced in the EV system did not show any correlation with the SSA of the n-ZnO activators (Figure 7A). However, there were more complex differences between the crosslink structures of the vulcanizates.

Generally, the application of n-ZnO raised the crosslink density of the SBR vulcanizates in comparison with those containing the micrometric activator. The highest crosslink density was observed for the vulcanizates containing type D or type C particles, with a rod-like or hybrid rod-like morphology (also rod-like, but with less dense packing). As expected, the EV sulfur curing system produced vulcanizates with shorter crosslinks [29], as can be seen in Figure 7B. With the exemption of the C-type particles, the application of n-ZnO greatly reduced the content of polysulfidic crosslinks in the rubber vulcanizates compared with the reference—the standard ZnO activator. This effect was the most visible in the case of the rod-like (B) and the hybrid (D) morphologies, despite their moderate SSA. The reduction in long crosslinks was in favor of short monosulfidic and C–C crosslinks.

### 3.4. Mechanical Properties of Rubber Vulcanizates

#### 3.4.1. Conventional Vulcanization System

The smaller amounts of stronger short crosslinks (C–C, mono-, and di-sulfidic) in the vulcanizates containing n-ZnO activators in comparison with the reference vulcanizate (with standard ZnO) resulted in a lower moduli and smaller durability in extension [30], as shown in Figure 8.

There was practically no difference between the mechanical properties of the vulcanizates prepared in the CV system with the n-ZnO activators of different morphologies. Irrespective of their morphology, SSA, crosslink density, and structure, the application of n-ZnO activators in the conventional sulfur curing system had an adverse effect on the mechanical properties of the rubber vulcanizates compared with the vulcanizates containing conventional micro-ZnO.

#### 3.4.2. Effective Vulcanization System

Compared with the reference vulcanizate containing standard ZnO, the vulcanizates containing n-ZnO activators had a lower content of stronger short crosslinks (with the exception of the EV_C sample). This resulted in their having a higher moduli [29] (with the exception of EV_G), but did not necessarily result in greater durability in extension, as shown in Figure 9.

The vulcanizates EV_D and EV_E showed the highest stiffness and durability. This seemed to have been as a result of their combination of high crosslink density and high content of short crosslinks. However, there was no correlation between the mechanical properties of the rubber vulcanizates and either the morphology or SSA of the n-ZnO particles. In contrast to the CV system, the incorporation of n-ZnO particles in the EV curing system resulted in changes to the crosslink structure, which affected the echanical properties of the vulcanizates. These results justify the application of ZnO nanopowders with a hybrid (D) or cauliflower (E) morphology.

## 4. Conclusions

Nano-powder particles of zinc oxide (n-ZnO) behave differently in sulfur crosslinking systems, depending on their morphology and SSA. Their performance also depends on the activity of the curing system used. The course and kinetics of vulcanization, as well as the crosslink density, structure, and mechanical properties of the vulcanizates, seem to depend more on the morphology of the n-ZnO particles than on their SSA. In this study, nanoparticles with a cauliflower morphology exhibited a higher squalene wettability than rod-like nanoparticles, irrespective of their SSA. However, the curing systems (especially EV) containing cauliflower-shaped n-ZnO required a higher activation energy. Both the activation energy and the speed of vulcanization (d*α*/d*t*) were found to depend on the morphology of the n-ZnO particles. In the CV system, vulcanization was faster when the n-ZnO particles with a rod-like morphology were used, compared with particles with a cauliflower morphology. The situation was the opposite in the case of the EV systems, in which the cauliflower-shaped particles sped up the vulcanization process. The reduction in activation energy was most visible in the case of EV systems, but again only when n-ZnO particles with a rod-like morphology were applied. In the EV system, the application of cauliflower structures (E and G), despite resulting in a higher activation energy, accelerated the process compared with the reference micro-ZnO powder. The crosslink density of the vulcanizates did not correlate with either the morphology or the SSA of the applied n-ZnO particles. However, a slight increase in the crosslink density was associated with cauliflower-shaped n-ZnO in the EV system. The morphology of the n-ZnO particles had practically no influence on the crosslink structure and related mechanical properties of the CV vulcanizates. However, rod-like nanoactivators used in the EV system resulted in a greater reduction in the content of polysulfidic crosslinks in comparison to the rubber vulcanizates containing cauliflower-shaped n-ZnO. This effect was the most pronounced in vulcanizates containing n-ZnO with a hybrid morphology (D). These vulcanizates showed both the highest crosslink density and the greatest reduction in polysulfidic crosslinks.

Based on the results of this study, the morphology of n-ZnO activator particles appears to influence the parameters and activation energy for vulcanization, as well as the crosslink density, structure, and mechanical properties of the rubber vulcanizates. This influence is independent of the SSA of the n-ZnO particles, but depends on the efficiency of the curing system.

## Figures and Tables

**Figure 1 polymers-13-01040-f001:**
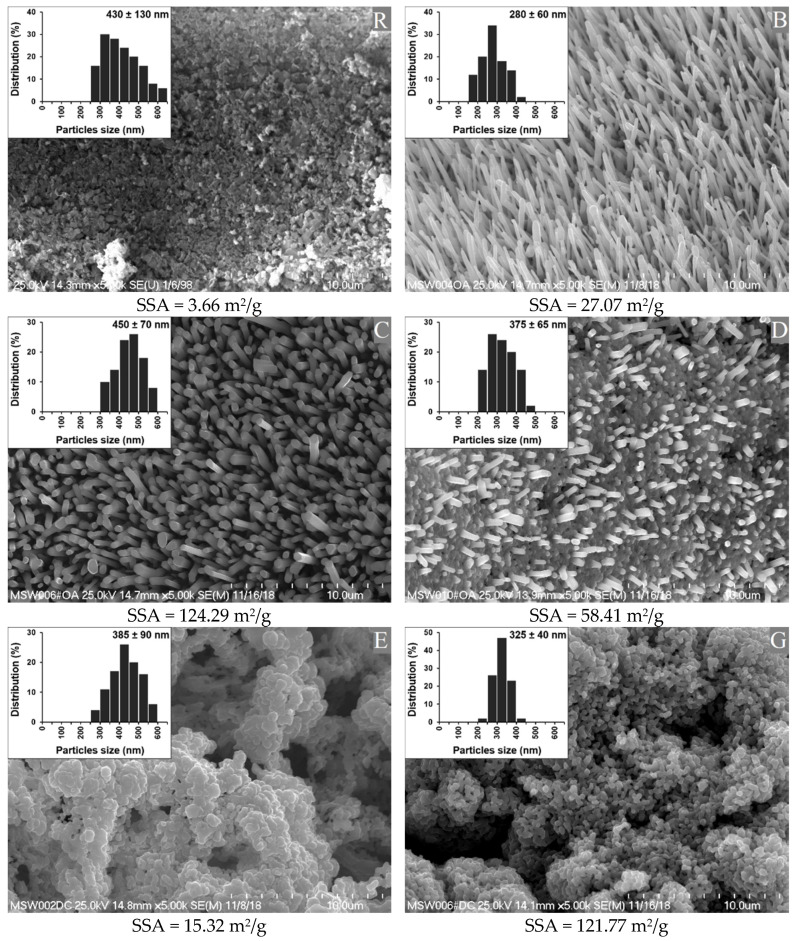
Morphology, specific surface area (SSA), and particle size of the reference (R) ZnO and the synthetized n-ZnO powders (B, C, D, E and G).

**Figure 2 polymers-13-01040-f002:**
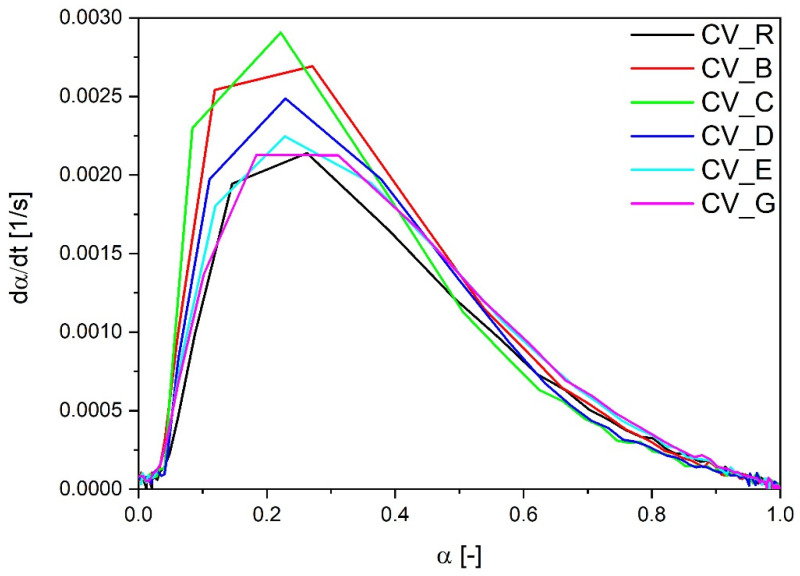
Vulcanization speed as a function of vulcanization degree for rubber compounds prepared in the CV system. CV_...—rubber compounds containing n-ZnO in a conventional curing system; CV_R—reference rubber compound containing standard ZnO in a conventional curing system.

**Figure 3 polymers-13-01040-f003:**
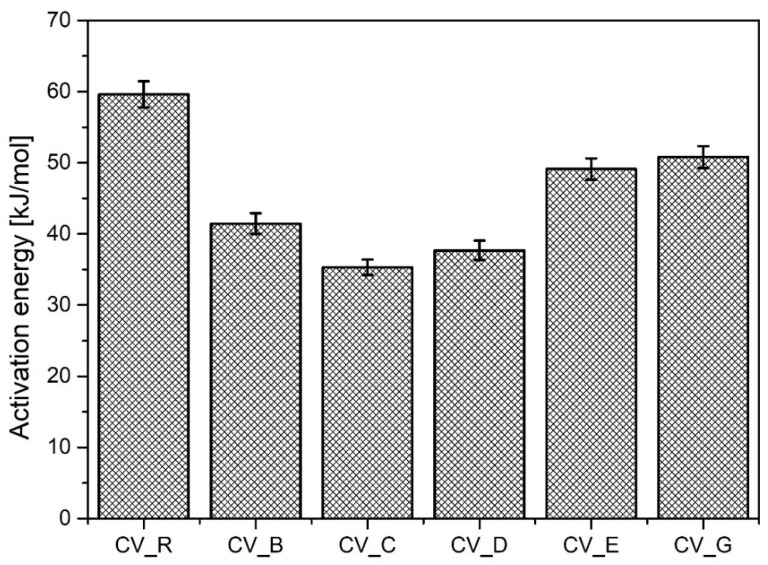
Activation energy for vulcanization of rubber compounds prepared in the CV system. CV_...—rubber compounds containing n-ZnO in a conventional curing system; CV_R—reference rubber compound containing standard ZnO in a conventional curing system.

**Figure 4 polymers-13-01040-f004:**
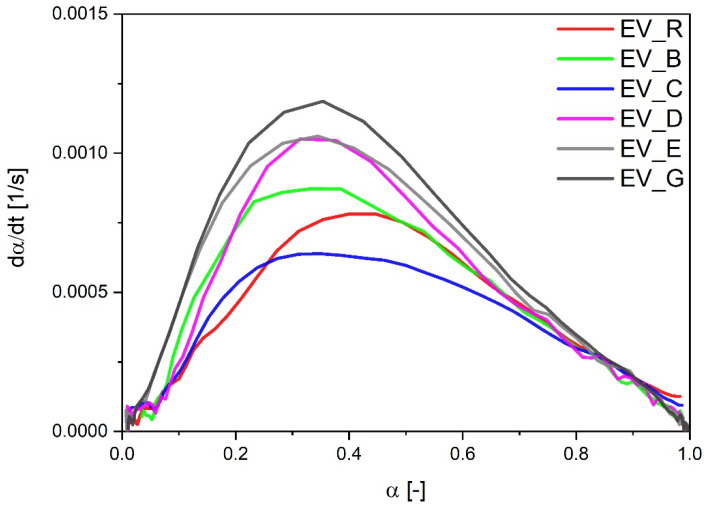
Vulcanization speed as a function of the vulcanization degree for rubber compounds prepared in the EV system. EV_...—rubber compounds containing n-ZnO in an effective curing system; EV_R—reference rubber compound containing standard ZnO in effective curing system.

**Figure 5 polymers-13-01040-f005:**
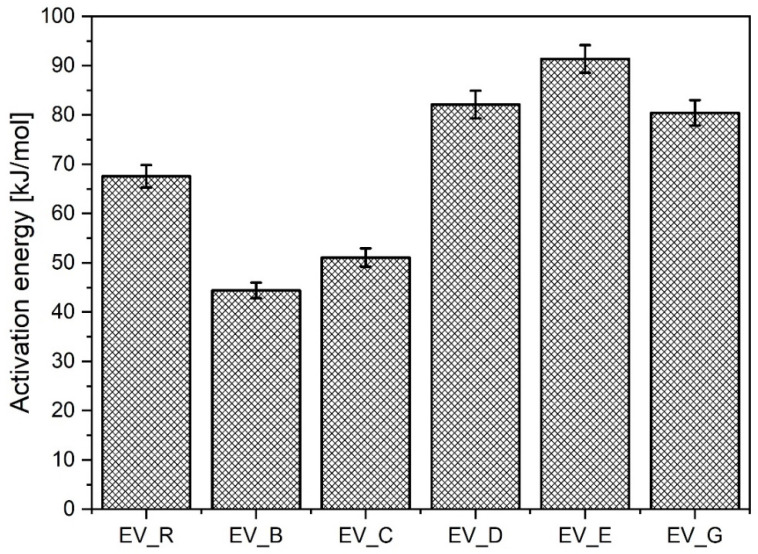
Activation energy for the vulcanization of rubber compounds prepared in the EV system. EV_...—rubber compounds containing n-ZnO in the effective curing system; EV_R—reference rubber compound containing standard ZnO in the effective curing system.

**Figure 6 polymers-13-01040-f006:**
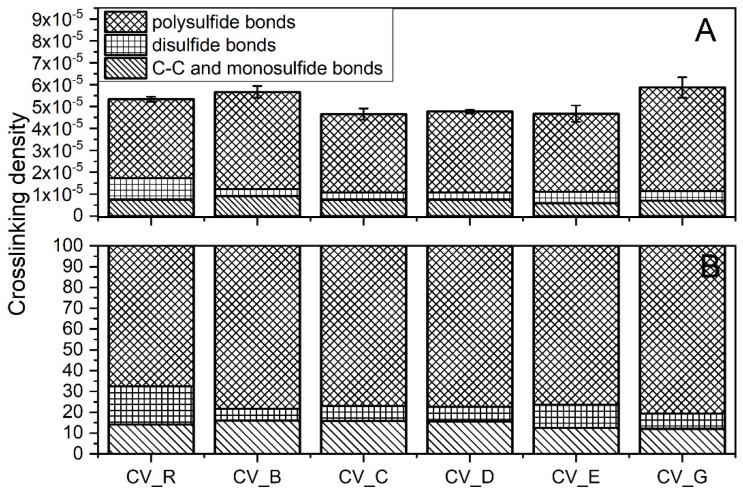
Crosslink structure of the rubber compounds prepared in the CV system: (**A**) content of crosslinks (mole/cm^3^); (**B**) division of crosslinks [%]. CV_...—rubber compounds containing n-ZnO in conventional curing system; CV_R—reference rubber compound containing the reference ZnO in a conventional curing system.

**Figure 7 polymers-13-01040-f007:**
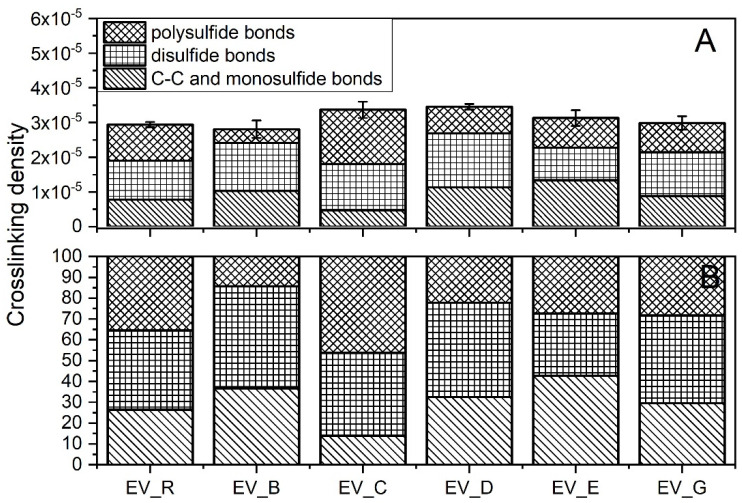
Crosslink structure of rubber compounds produced in the EV system: (**A**) content of crosslinks (mole/cm^3^); (**B**) division of crosslinks (%). EV_...—rubber compounds containing n-ZnO in the effective curing system; EV_R—reference rubber compound containing standard ZnO in the effective curing system.

**Figure 8 polymers-13-01040-f008:**
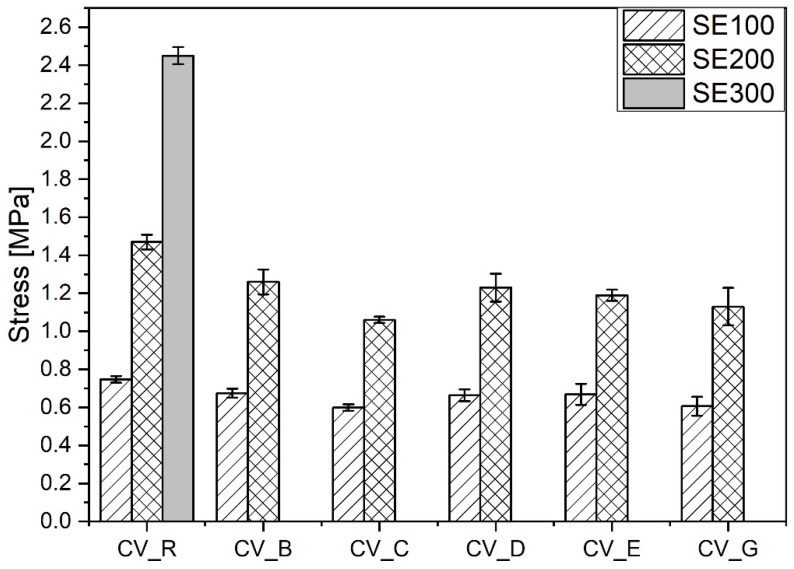
Mechanical properties in the extension of rubber compounds cured in the CV system. CV_...—rubber compounds containing n-ZnO in a conventional curing system; CV_R—reference rubber compound containing standard ZnO in a conventional curing system.

**Figure 9 polymers-13-01040-f009:**
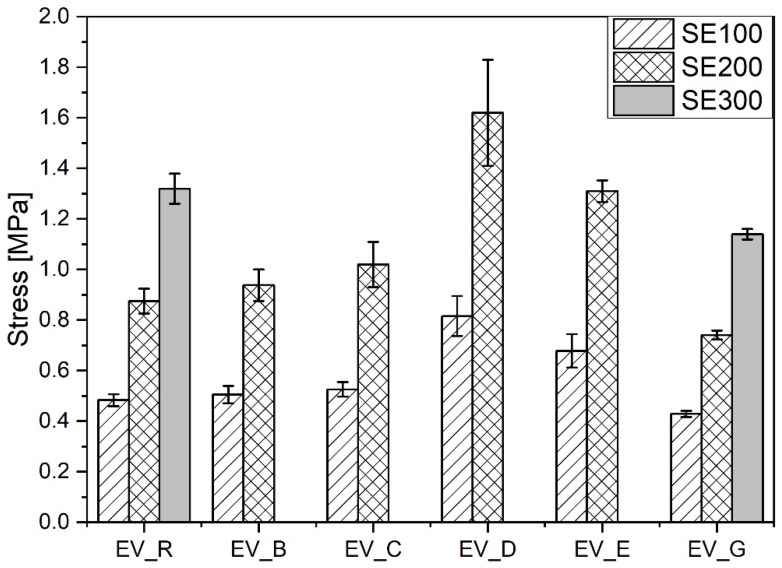
Mechanical properties in the extension of rubber compounds cured in the EV system. EV_...—rubber compounds containing n-ZnO in the effective curing system; EV_R—reference rubber compound containing the standard ZnO in the effective curing system.

**Table 1 polymers-13-01040-t001:** Classification of accelerator groups and their relative curing speeds [2].

Type	Abbreviation	Relative Curing Speed
Guanidines	DPG	slow
Dithiocarbamates	ZDBC	very fast
Thiurams	TMTD, TMTM and DPTTS	very fast
Thioureas	ETU	fast
Thiophosphates	DIPDIS	semi-fast
Thiazoles	MBT, MBTS and ZMBT	moderate

**Table 2 polymers-13-01040-t002:** Components used in conventional (CV), semi-efficient (semi-EV), and efficient (EV) vulcanization systems [2].

Type	Sulfur (*S*) [phr]	Accelerator (*A*) [phr]	*A*/*S*
CV	2–3.5	1.2–0.4	0.1–0.6
Semi-EV	1–1.7	2.5–1.2	0.7–2.5
EV	0.4–0.8	5–2	2.5–12

**Table 3 polymers-13-01040-t003:** Components used in CV, semi-EV, and EV vulcanization systems [2].

	Curing System	Sulfur (*S*) [phr]	Accelerator (*A*) [phr]	A/S
Crosslink Type				
poly- and disulfidic		95	50	20
monosulfidic and C–C		5	50	80
cyclic sulfide conc.		high	medium	low

**Table 4 polymers-13-01040-t004:** Types of precursors and the parameters for n-ZnO synthesis [13,14].

Kind of n-ZnO	Precursor Used
B ^1^	[Zn(CH_3_COO)_2_(hmta)(H_2_O)]
C ^1^, G ^2^	[Zn_3_(OH)(C_2_H_5_COO)_5_(hmta)]_n_
D ^1^	[Zn_2_(i-C_3_H_7_COO)_4_(hmta)_2_]
E ^2^	[Zn(H_2_O)_6_]_2_^+^·2(HCOO-)·2(hmta)·4H_2_O

^1^ Precursor dispersed in oleic acid using ultrasound, then heated at 220 °C for 2 h. After this time, the temperature was increased to 800 °C at the maximum heating rate (30 °C/min) to purify it of oleic acid. ^2^ Precursor heated at a rate of 5 °C/min to 500 °C, then left for 1 h at this temperature.

**Table 5 polymers-13-01040-t005:** Composition (phr) of the rubber compounds.

Components	Conventional System (CV)	Effective System (EV)
Buna SL 4525-0 (s-SBR)	100	
sulfur	3	0.6
n-tert-butyl-2-benzothiazole sulfenamide (TBBS)	1.5	4
Stearic acid	3	3
n-ZnO	1.5	1.5

**Table 6 polymers-13-01040-t006:** Squalene sorption line slopes for the different ZnO particles.

n-ZnO Type	Sorption Line Slope [g^2^/s × 10^−3^]
R	1.951
B	0.894
C	1.637
D	0.464
E	2.722
G	3.963

**Table 7 polymers-13-01040-t007:** Parameters for the vulcanization of rubber compounds produced in the CV system.

	Parameter	*M*_L_ (dNm)	*M*_H_ (dNm)	∆*M* (dNm)	*t*s_2_ (min)	*t*_90_ (min)	*CRI* (%/min)
Sample							
CV_R		0.6	6.3	5.7	17.7	34.1	6.1
CV_B		0.5	6.9	6.4	12.2	28.8	6.0
CV_C		0.5	6.6	6.1	12.5	32.3	5.1
CV_D		0.5	6.1	5.6	13.9	32. 8	5.3
CV_E		0.5	6.9	6.4	12.7	27.9	6.5
CV_G		0.5	6.8	6.3	13.1	27.8	6.8

**Table 8 polymers-13-01040-t008:** Parameters for the vulcanization of rubber compounds prepared in the EV system.

	Parameter	*M*_L_ [dNm]	*M*_H_ [dNm]	∆*M* [dNm]	*t*s_2_ [min]	*t*_90_ [min]	*CRI* [%/min]
Sample							
EV_R		0.5	4.1	3.6	34.4	48.9	6.9
EV_B		0.6	4.2	3.6	23.7	39.1	6.5
EV_C		0.5	4.0	3.5	32.2	47.3	6.6
EV_D		0.4	4.2	3.8	16.9	30.2	7.5
EV_E		0.6	4.4	3.8	22.9	37.3	7.0
EV_G		0.6	4.5	3.8	22.5	35.9	7.5

## Data Availability

The raw/processed data required to reproduce these findings cannot be shared at this time due to technical or time limitations.

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
