# Peer review of "Influence of n-ZnO Morphology on Sulfur Crosslinking and Properties of Styrene-Butadiene Rubber Vulcanizates"

_polymers, 2021, doi:10.3390/polym13071040_

Round 1

Reviewer 1 Report

The submitted manuscript concerns itself with quantifying the effect (on sulfur crosslinking) of adding n-ZnO particles to styrene-butadiene rubber.

The introduction of the article contains a comprehensive exposition in which the first part details the contents of classes of materials, while the second part (2 paragraphs) describe the state of the art, with several good references. The introduction ends with the mention of the main goal of the paper - increasing the understanding of the effect of the shape of n-ZnO particles on curing activity, but also the influence of the particle shape on the crosslink structures. The SEM images for the different samples with various specific surface areas are interesting on thier own.

On of the main points of the paper is that the morphology and specific surface area of the zinc oxide nanoparticles have a significant influence. I believe with the data presented, this is clearly demonstrated, although this behaviour was to be expected. In addition, a clear influence on the kinetics is also shown and the authors make a few interesting remarks regarding the effect of different particle shapes with the same surface area, especially regarding wettability.

It is interesthing though that these two aspects (morphology and specific surface area of particles) seem to not have a direct effect in crosslink density. Perhaps the absence of correlation could be a explained a bit? Or make at least a hypothesis for why this is the case.

Overall, I must say that the authors' attention to details is a strong point of the paper. Firstly, the work was done very meticulously, all the data regarding materials and methods are very clearly specified, which shoud insure reproduciblity of results, which may be a problem in these types of experiments. Also, the authors have done a very good job on providing a wealth of observations and explanations, such that this paper isn't simply a characterization of a new material, but an thorough in-depth analysis which uses several methods, and is firmly based in theory. 

Therefore, I warmly encourage the publication of the article in Polymers. 

Author Response

Thank you very much for your kind opinion about our paper. Your question concerning aspects of morphology and specific surface area of n-ZnO particles indeed seem to not have a direct effect in crosslink density. We try to explain a bit the absence of the correlation, based on the available literature. See the manuscript for details.

Reviewer 2 Report

The present manuscript deal with the study of the effect of different structure of zinc oxide nanoparticles (ZnO) on the activation energy, vulcanization parameters, crosslink density, crosslink structure, and mechanical properties in extension of sulfur vulcanizates of styrene-butadiene rubber (SBR).

Mayor revision:

  1. Please specify how you prepare the sample for SEM measurement.
  2. How many sample of each investigated type of ZnO you use for performed mechanical properties. Please specify this date in Experimental Part.
  3. In the Figure 1, the image size for each simple is different please try to use the same images size for different nanoparticles type.

Minor revision:

  1. Please use “SEM image” instead of “SEM pictures”. Please revise whole manuscript.
  2. In the Figure 1 please use alphabetic order to described each SEM image (A, B, C, D, E, F, G). The R should be A, ect.
  3. Please extend the legend of the Figure 1, identify A, B, C, etc.
  4. Please specify the size of the nanoparticles for each type? Do you find different size for each type? Do you think that the size of the nanoparticles can be affect BET and swelling measurement? Please try to related this point in your manuscript.
  5. in the legend of the Figure 3 please specify what CV_R, CV_B, CV_C etc mean. Please remember to change R for A since this is now reason for named A as R.

Author Response

Thank you very much for your effort in reviewing our work. We try to address all your remarks, hoping our answers and explanations meet your requirements.

Major revision:

  1. The methodology of preparing ZnO powders for SEM mesurements has been specified. See the manuscript.
  2. The number of each investigated type of ZnO used in the mechanical tests has been specified. See the manuscript for details.
  3. SEM images in the Figure 1 have been unified and additionally provided with the information on the particle size distribution for each ZnO sample (see the minor revision no. 4).

Minor revision:

  1. “SEM pictures” has been replaced by “SEM image” in the whole manuscript.
  2. We decided to leave the names of ZnO samples, because in the case of synthetized n-ZnO particles they follow the description used in our previous publications [13, 14] for their precursors, as given in the Table 4. In this case the R should be also left unchanged for the reference ZnO sample in our opinion.
  3. The legend of the Figure 1 has been modified. All required information concerning the powders has been provided with the SEM images. The remark concerning the symbols of the samples, has been replied above
  4. See the revision made to the Figure 1. We agree with the Reviewer that the size and morphology of the nanoparticles can affect BET and swelling measurement. The remark has been addressed in the manuscript and referred to the literature [23].
  5. The legend for the Figure 3 has been made more detailed as requested. To be consistent the legends of the rest of the Figures have been completed accordingly.

Round 2

Reviewer 2 Report

Accept in present form